# Treatment options and barriers to case management of neonatal pneumonia in India: a protocol for a scoping review

N Sreekumaran Nair,[1] Leslie Edward Lewis,[2] Shruti Murthy,[3] Myron Anthony Godinho,[3] Theophilus Lakiang,[3] Bhumika T Venkatesh[3]

## ABSTRACT

**Introduction** India contributes to the highest neonatal deaths globally. Case management is said to be the cornerstone of pneumonia control. Much of the published evidence focuses on children aged 1 to 59 months. This scoping review, thus, aims to identify the treatment options for and barriers to case management of neonatal pneumonia in India.

**Methods and analysis** This protocol is part of a series of three reviews on neonatal pneumonia in India. Studies addressing treatment of or barriers to case management of neonatal pneumonia in Indian context, published in English in peer-reviewed and indexed journals will be eligible for inclusion. Electronic search will be conducted on nine databases. Hand searching and snowballing will be done for published and grey literature. Selection of studies will be done in title, abstract and full-text stages. A narrative summary will be performed to summarise the details of evidence.

**Ethics and dissemination** As this is a review involving analysis of secondary data which is available in the public domain and does not involve human participants, ethical approval was not required. The findings of the study will be shared with all stakeholders of this research. Knowledge dissemination workshops will be conducted with relevant stakeholders to ultimately transfer the evidence tailored to the stakeholder (eg, policy briefs, publications, information booklets and so on).

**PROSPERO 2016** CRD42016045449

[1]Department of Statistics, Manipal University, Manipal, India
[2]Department of Pediatrics, Kasturba Medical College, Manipal University, Manipal, India
[3]Public Health Evidence South Asia (PHESA), Manipal University, Manipal, India

**Correspondence to**
Dr N Sreekumaran Nair; sree.nair@manipal.edu

## Strengths and limitations of this study

► Most documented literature in pneumonia case management addresses postneonatal age groups. First review to consolidate research on the treatment modalities, their implementation and the barriers to the case management of neonatal pneumonia in India.

► A comprehensive search strategy was developed over nine databases including relevant regional databases and grey literature published in any year.

► The review will narratively summarise published literature. No quality assessment of included studies is planned.

## INTRODUCTION

Globally, 5.9 million children died in 2015, out of which 2.6 million were neonates.[1] India accounts for more neonatal deaths than any other country.[2] More than half of the child deaths from pneumonia occur during the newborn period.[3] In India, economic deprivation, impaired access to healthcare, harmful child-rearing practices, malnutrition and indoor air pollution are all major risk factors for pneumonia.[4] [5] Neonatal pneumonia is particularly difficult to define and classify, as witnessed by differing definitions in different studies.[3] [6]

In 2008, the WHO-United Nations Children's Fund formulated the Global Action Plan for the Prevention and Control of Pneumonia (GAPP),[7] and in India, the Integrated Management of Neonatal & Childhood Illness (IMNCI) is now an important part of the national strategy to control childhood illness. Timely detection, effective case management and prompt referral can reduce child morbidity and mortality due to pneumonia.[8] However, this is challenging in regions where comorbid conditions (eg, tuberculosis, malaria and AIDS) and antibiotic resistance prevail.[6] These difficulties are compounded by the clinical overlap of neonatal sepsis and pneumonia,[3] and obscured by conditions like hyaline membrane disease which mimic neonatal pneumonia and impede detection in the absence of bacteriological confirmation.[5] Moreover, injudicious antibiotic therapy could lead to resistance or treatment failure.[9] A multitude of factors such as these pose special challenges to the initiation and maintenance of treatment, resulting in excessive morbidity and mortality.[8] However, most documented difficulties in pneumonia case management address post-neonatal age groups and fail to discuss neonatal treatment issues.

There is a recognised need for consolidated research on the treatment modalities, their implementation and the barriers to the

case management of neonatal pneumonia in India. This evidence is required to inform the development of interventions, education and preventive strategies to combat this scourge of India's newborn. Thus, this scoping review will attempt to synthesise evidence on different treatment options existing for neonatal pneumonia and the factors hindering effective case management of neonatal pneumonia in the Indian context.

The objectives of this scoping review are to identify:
1. Treatment options for neonatal pneumonia.
2. Barriers to case management of neonatal pneumonia in the Indian context.

This protocol is part of a larger mixed-methods research project consisting of a qualitative study and a trilogy of two systematic and one scoping reviews on neonatal pneumonia in India addressing risk factors, management and predictors of mortality due to neonatal pneumonia in the Indian context.

## METHODS AND ANALYSIS
This review will be conducted from August 2016 to October 2017.

### Criteria for considering studies for this review
#### Types of studies
##### Inclusion criteria
Studies eligible for inclusion should have been conducted among neonates with pneumonia (or sepsis) in the Indian context and their stakeholders. Primary studies (of any study design including editorials, case reports, case series, cross-sectional studies, case control studies, cohort studies, intervention studies and qualitative studies), policy papers, guidelines, reports and fact sheets addressing treatment of or barriers to case management of neonatal pneumonia in Indian context will be eligible to be included in the review. Studies have to be published in English language in indexed and peer-reviewed journals to be eligible for inclusion.

##### Exclusion criteria
The following studies will be excluded: all types of reviews, meta-analysis, conference papers and reports which do not report on treatment or barriers to case management of neonatal pneumonia in an Indian context.

Operational definitions: for the purpose of this review, treatment was operationally defined as 'any specific or supportive treatment administered to a neonate with pneumonia'; case management was defined as 'detection, investigation, treatment, referral, monitoring, support or follow-up of pneumonia in a neonate either in the facility or community'[8]; and barrier to case management was defined as 'any difficulty or obstacle during the case management of neonatal pneumonia'. Though we defined these terms in the beginning, our intention will also be to capture the definitions, where available, as reported by the authors and present them in the narrative synthesis.

### Type of participants
Neonates with pneumonia in Indian context.

### Outcomes of this review
Outcomes of this review will be (a) specific and supportive treatment and (b) barriers to case management of neonatal pneumonia in Indian context.

### Search methods for identification of studies
A comprehensive and relevant search strategy to identify all relevant studies will be developed by reviewing literature and discussion with subject experts and an information scientist. The search terms used and search strategies for PubMed have been provided in tables 1 and 2.

| Table 1 | Search strategy for treatment options (PubMed) |
|---|---|
| **Strategy: #1 AND #2 AND #3** | |
| #1 | **(((Neonate\* OR childhood OR neonatal\* OR newborn\* OR 'young infant' OR child OR paediatric OR pediatric\* OR 'neonatal period' OR infant\* OR 'newborn infant')))** |
| #2 | ((((((((((((((((((((((((Pneumonia\*) OR Pneumon\*) OR 'community acquired pneumonia') OR 'congenital pneumonia') OR 'hospital acquired pneumonia') OR 'nosocomial pneumonia') OR 'ventilator associated pneumonia') OR 'early onset pneumonia') OR 'late onset pneumonia') OR 'infective pneumonia') OR 'infectious pneumonia') OR 'meconium aspiration syndrome') OR 'meconium aspiration') OR 'lipoid pneumonia') OR sepsis\*) OR 'acute respiratory infections') OR 'early onset sepsis') OR 'chemical pneumonia') OR 'aspiration pneumonia') OR 'late onset sepsis') OR infection\*) OR 'nosocomial infection') OR 'early onset infection') OR 'late onset infection') OR 'acute lower respiratory infection') OR 'hospital acquired infection') OR 'congenital infection') OR 'viral pneumonia') OR 'gastro esophageal reflux disease') OR 'cystic fibrosis') |
| #3 | (((Treatment\* OR Therap\* OR 'Patient care management' OR 'Case management programs' OR 'Home based neonatal care' OR 'Case Management' OR 'Clinical case management' OR 'Community case management' OR 'Integrated community case management' OR 'Home based newborn care' OR 'Case management models' OR Antibiotic\* OR Ventilation\* OR 'Intensive care units' OR 'Intensive care' OR 'Neonatal intensive care units' OR 'Special Newborn Care Units' OR 'Injectable antibiotic' OR 'oral antibiotic' OR 'supportive therapy' OR 'specific therapy' OR 'specific treatment' OR 'Supportive treatment')))) |

Geographical filter: India
Language Filter: English

| Table 2 | Search strategy for barriers to case management (PubMED) |
|---|---|
| **Strategy: #1 AND #2 AND #3 AND #4** | |
| #1 | ((((Neonate* OR childhood OR neonatal* OR newborn* OR 'young infant' OR child OR paediatric OR pediatric* OR 'neonatal period' OR infant* OR 'newborn infant'))) |
| #2 | (((((((((((((((((((((((((((Pneumonia*) OR Pneumon*) OR 'community acquired pneumonia') OR 'congenital pneumonia') OR 'hospital acquired pneumonia') OR 'nosocomial pneumonia') OR 'ventilator associated pneumonia') OR 'early onset pneumonia') OR 'late onset pneumonia') OR 'infective pneumonia') OR 'infectious pneumonia') OR 'meconium aspiration syndrome') OR 'meconium aspiration') OR 'lipoid pneumonia') OR sepsis*) OR 'acute respiratory infections') OR 'early onset sepsis') OR 'chemical pneumonia') OR 'aspiration pneumonia') OR 'late onset sepsis') OR infection*) OR 'nosocomial infection') OR 'early onset infection') OR 'late onset infection') OR 'acute lower respiratory infection') OR 'hospital acquired infection') OR 'congenital infection') OR 'viral pneumonia') OR 'gastro esophageal reflux disease') OR 'cystic fibrosis') |
| #3 | ((Treatment* OR Therap* OR 'Patient care management' OR 'Case management programs' OR 'Home based neonatal care' OR 'Case Management' OR 'Clinical case management' OR 'Community case management' OR 'Integrated community case management' OR 'Home based newborn care' OR 'Case management models' OR Antibiotic* OR Ventilation* OR 'Intensive care units' OR 'Intensive care' OR 'Neonatal intensive care units' OR 'Special Newborn Care Units' OR 'Injectable antibiotic' OR 'oral antibiotic' OR 'supportive therapy' OR 'specific therapy' OR 'specific treatment' OR 'Supportive treatment')) |
| #4 | (((Barriers* OR challenging OR challenge* OR obstacle* OR difficult* OR drawback OR problem* OR hurdle* OR hindrance* OR hinder* OR gap* OR cost* OR utilization OR satisfaction))))) |

Geographical filter: India.
Language filter: English.

## Electronic searches

We will search PubMed, Ovid Medline, ProQuest, EMBASE, CINAHL, Web of Science, SCOPUS, WHOLIS and IndMED.

## Hand searching

Hand searching will be conducted for reports/guidelines/journal volumes not included in electronic databases and conference proceedings to review the references and contact the authors for full text of identified literature.

## Searching the grey literature

Potential sources of grey literature will include Shodhganga (INFLIBNET) and Government of India databases for reports, fact sheets and guidelines/policies in the Indian context.

## Reference lists

Snowballing will be conducted to screen the references of identified literature for potentially relevant studies.

## Data collection and management

The results (titles and/or abstracts) of the search will be managed using Endnote (v. x7). Study selection will be performed on Endnote (v. x7). Data will be extracted on Microsoft Excel 2007.

## Selection of studies

Studies will be reviewed based on the exclusion and inclusion criteria by two authors (SM and TL) independently in three stages. Stage I (title screening) will include assessment of each title for inclusion in the review. If both authors reject a title, it will not be included in the review. Studies which are approved by either author will move to the second stage of appraisal. Stage II (abstract screening) will involve screening of abstracts of the titles selected in stage I for inclusion in the review. If both authors reject an abstract at this stage, it will not be included in the review. Studies which are approved by either author will move to the third and final stage of appraisal. Stage III, the full-text screening stage, will comprise screening the full text of the abstracts selected in stage II. Only those studies approved by both the authors at this stage will be included in the review. In the event of a study being accepted by one author and rejected by another, a third author (MG) and a senior reviewer (SN or LL) will arbitrate and a consensus will be reached on whether to include the study or not.

## Data extraction and charting the results

A charting form was developed, in Microsoft Excel 2007, through an iterative process involving discussions and pilot testing. After a round of discussion among the authors, senior reviewers, subject and clinical experts, and statisticians, the form was pilot-tested on one study of each study design to ensure that it adequately facilitated the collection of essential information required for the narrative synthesis. The key headings under which charting will be done include (1) study characteristics, (2) methodological characteristics, (3) treatment options and barriers to case management and (4) other important information.

This standardised, pretested charting form will be used independently by two authors (SM and TL) to extract data from the selected studies. Disagreements will be resolved in the presence of the third (MG) and senior review authors (SN and LL) by discussion and consensus. Any discrepancies regarding inclusion of the study in the review will be discussed with the team and advisory group, and a decision will be made regarding its inclusion in the review.

## Dealing with missing data

In case of inadequacy of data, missing information, lack of clarity on information in methodology, or if outcomes are missing, authors of the respective studies will be contacted in an attempt to obtain the required details. Despite this attempt, if the missing data retrieval on some aspects of the outcome (like clarity and inadequacy) is not possible, the study will be included in the narrative summary with a mention of the same.

## Reporting the results

The complete results of any analyses conducted, including the final search strategy, will be reported. Results will be in tabular form supplemented with a descriptive summary of the findings. Tables will present the characteristics of included studies (study ID, year of publication, location and setting, study design and sample size, definitions adopted in the studies, treatments recommended by guidelines, treatments reported by primary research studies for neonatal pneumonia and barriers reported during the case management of neonatal pneumonia). The descriptive summary will include details about the study objectives, the approach adopted and the findings. A discussion, where applicable, on study limitations that should be considered when interpreting the findings of the review will be included. No quality assessment of the included studies has been planned.

A "Preferred Reporting Items for sSstematic Reviews and Meta-Analyses" (PRISMA) chart will be created to outline and summarise this study selection process.[10] The findings of this review will be reported in accordance with the 'guidance for conducting systematic scoping reviews'.[11]

**Acknowledgements** The authors would like to thank the following individuals for their continuous support and guidance during this process of protocol development: Dr Manoj Das, Director Projects, The INCLEN Trust International, New Delhi; Dr Anju Sinha, Deputy Director General, Scientist 'E', Division of Child Health, Indian Council of Medical Research, New Delhi; Dr KK Diwakar, Professor and Head, Department of Neonatology, Associate Dean, Malankara Orthodox Syrian Church Medical College, Kerala; Mrs Ratheebhai V, Senior Librarian and Information Scientist, at Manipal School at Communication, Manipal University, Manipal; Dr Ravinder M Pandey, Professor and Head, Department of Biostatistics, All India Institute of Medical Sciences, New Delhi; Dr B Shantharam Baliga, Professor, Department of Paediatrics, Kasturba Medical College, Mangalore, Karnataka,; Dr Shirish Darak, Senior Researcher, PRAYAS, Pune, Maharashtra; Dr Unnikrishnan B, Associate Dean and Professor, Department of Community Medicine, Kasturba Medical College, Mangalore. They also thank Public Health Evidence South Asia (PHESA) and Manipal University, Manipal for providing the necessary institutional and infrastructural support for the project. They would also like to thank The INCLEN Trust International, New Delhi, and The Bill and Melinda Gates Foundation for the financial support which made this project possible.

**Contributors** NSN: guarantor of the review. NSN, BTV and LEL: conceived the research idea and reviewed the manuscript. NSN and LEL: provided overall technical guidance. LEL: assisted in developing search terms. SM, TL and MG: designed the protocol, drafted the manuscript and developed and pilot tested the search strategies and data extraction form.

**Funding** This project is supported by a grant from Bill and Melinda Gates Foundation (grant OPP1084307) to The INCLEN Trust International and sub-grant to Manipal University (subgrant INC2015GNT004). The views expressed through this project do not necessarily represent the views of Bill and Melinda Gates Foundation or The INCLEN Trust International or Manipal University.

**Competing interests** All authors have completed the ICMJE uniform disclosure form atwww.icmje.org/coi_disclosure.pdf and declare: all authors had financial support (grants) from Bill and Melinda Gates Foundation (grant OPP1084307) to The INCLEN Trust International and subgrant to Manipal University (subgrant INC2015GNT004)., during the conduct of the study and for the submitted work; no financial relationships with any organisations that might have an interest in the submitted work in the previous 3 years; no other relationships or activities that could appear to have influenced the submitted work.

**Provenance and peer review** Not commissioned; externally peer reviewed.

**Data sharing statement** All data supporting this study will be provided as supplementary material together with the manuscript of the study's final results.

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
