## [Reviewer comments · BMJ Open]

ARTICLE DETAILS

TITLE (PROVISIONAL)	Treatment options and Barriers to Case Management of Neonatal Pneumonia in India: a protocol for a scoping review
AUTHORS	Sreekumaran Nair, N; Lewis, Leslie; Murthy, Shruti; Godinho, Myron; Lakiang, Theophilus; Venkatesh, Bhumika

VERSION 1 - REVIEW

REVIEWER	Denny John Campbell Collaboration, New Delhi I am on the Doctoral Advisory Committee of Dr. Shruti Murthy, however this study is part of her PhD.
REVIEW RETURNED	31-May-2017

GENERAL COMMENTS	The reviewer also provided a marked copy with additional comments. Please contact the publisher for full details.
---

REVIEWER	Sarah Prentice London School of Hygiene and Tropical Medicine, United Kingdom
REVIEW RETURNED	05-Jun-2017

GENERAL COMMENTS	Thank you for requesting my review of this protocol for a systematic review. I believe that the questions that this review seeks to address are important, but I am concerned that the methods as presented will be insufficient to fully address them. My main concern is about the lack of any quality assessment of the included studies. If these results are intended to inform policy or protocols then I feel that some estimation of the quality of the studies included is vital (particularly for the treatment element of this study). You search #2 should also include alternative spellings of pediatric e.g paediatric.
---

REVIEWER	SRINIVAS MURKI FERNANDEZ HOSPITAL HYDERGUDA HYDERABAD TELANGANA INDIA
REVIEW RETURNED	10-Jun-2017

GENERAL COMMENTS	The statistics aspects of the study is missing. Details on the methods used for statistical analysis is not reported in the study. This needs to be addressed
---

VERSION 1 – AUTHOR RESPONSE

Reviewer: 1

1. Operational definitions: Barrier to case management in the context of India would include challenges due to costs of care, i.e. demand-side factors. Also, the issue with supply-side factors. The authors need to elaborate the definition to include issues around service use, caregiver outcomes, cost-effectiveness as well.

Response: The potential barriers to case management are wide and varied and exist in every area of healthcare service delivery. Our definition attempted to provide a general idea without restricting ourselves to any specific barriers.

2. Search methods for identification of studies: Authors mentioned secondary data analysis. I presume this will be public databases such as NSSO etc. This is not mentioned in the search methods. Also, these databases might not report costs of neonates with pneumonia directly. Authors will need to conduct the analysis of this data in addition.

Response: Thank you for your observation. We have revised this section to be more specific and it now reads “Primary studies, (of any study design including editorials, case reports, case series, cross-sectional studies, case control studies, cohort studies, intervention studies and qualitative studies), policy papers, guidelines, reports, and fact sheets, addressing treatment of or barriers to case management of neonatal pneumonia in Indian context were eligible to be included in the review.”

3. Search terms: Barriers related to cost are missing in the search strategy. Also, issues around service use, utilization, satisfaction, and cost effectiveness

Response: These words have now been included in the strategy.

4. Data synthesis: Protocol mentions that both quantitative and qualitative studies will be searched. Hence this is a mixed methods review. It is important to highlight this here. Also, the potential methods of mixed methods synthesis for synthesising results on patterns in the relationships between barriers and outcomes will also need to be mentioned.

Response: The objective of this review is not to synthesize the results, but to identify and compile a list of treatment options and barriers to case management. Though this review involves both study designs, this scoping review does not intend to collect and synthesize any “quantitative data” about the treatments and barriers to case management. Hence there is no scope for mixed-methods synthesis in this review.

5. Critical appraisal section missing. There is potential to use Mixed Methods Appraisal Tool (MMAT) which has been validated for the critical appraisal of studies with diverse designs. Also, use of interrater reliability for independent assessments. Use of sensitivity analysis for assessing the impact of lower quality studies.

Response: As indicated above, there is no intention of data synthesis and sensitivity analysis. While we plan to conduct inter-reliability assessment, we plan to conduct it and include it in the report of the project as a whole (which consists of three reviews).

Reviewer: 2

1. I believe that the questions that this review seeks to address are important, but I am concerned that the methods as presented will be insufficient to fully address them. My main concern is about the lack of any quality assessment of the included studies. If these results are intended to inform policy or protocols then I feel that some estimation of the quality of the studies included is vital (particularly for the treatment element of this study).

Response: The intent of this review is to identify and compile a list of treatment options and barriers to case management. This is meant to be conducted as a scoping review, and hence does not contain the components of quality assessment. We have amended the title to now identify the review as a 'scoping review.

2. You search #2 should also include alternative spellings of pediatric e.g paediatric.

Response: Alternate spellings of pediatric have been included on page 6 (Tables 1 and 2) of the manuscript.

Reviewer: 3

1. The statistics aspects of the study is missing. Details on the methods used for statistical analysis is not reported in the study. This needs to be addressed

Response: The intent of this review is to identify and compile a list of treatment options and barriers to case management. There is no scope for numerical data synthesis. Results will be described in narrative synthesis to summarize the details of evidence. A discussion, where applicable, on study limitations that should be considered when interpreting the findings of the review will be included. The complete results of any analyses conducted, including the final search strategy, will be reported.

VERSION 2 – REVIEW

REVIEWER	Denny John Campbell Collaboration, New Delhi, India
REVIEW RETURNED	30-Jun-2017

GENERAL COMMENTS	Since the title has been changed to be a scoping review the authors would need to refer to the guidelines for conducting systematic scoping reviews. One reference is here: Guidance for conducting systematic scoping reviews. Guidance for conducting systematic scoping reviews. International Journal of Evidence-Based Healthcare. 13(3):141–146, SEP 2015. Micah D.J. Peters; Christina M. Godfrey, DOI: 10.1097/XEB.0000000000000050
---

REVIEWER	Sarah Prentice LSHTM UK
REVIEW RETURNED	LSHTM UK

GENERAL COMMENTS	The authors have responded to my concerns and explained their methods more clearly in the revised manuscript. It is therefore a technically acceptable paper. However, I'm just not sure whether the results of this study would be of interest to many readers without an assessment of the quality of the studies they are reviewing (just listing possible available treatments and barriers is not that useful if some assessment of the extent to which they would be beneficial/a hinderance is not made at the sametime).
--

REVIEWER	SRINIVAS MURKI Fernanadez Hospital, Hyderguda, Hyderabad, India
REVIEW RETURNED	22-Jun-2017

GENERAL COMMENTS	No more corrections needed
----------------------------

VERSION 2 – AUTHOR RESPONSE

Reviewer: 3

Reviewer Name: SRINIVAS MURKI

Institution and Country: Fernanadez Hospital, Hyderguda, Hyderabad, India

Please state any competing interests or state 'None declared': NOne

Please leave your comments for the authors below

No more corrections needed

Reviewer: 1

Reviewer Name: Denny John

Institution and Country: Campbell Collaboration, New Delhi, India

Please state any competing interests or state 'None declared': None declared

Please leave your comments for the authors below

Since the title has been changed to be a scoping review the authors would need to refer to the guidelines for conducting systematic scoping reviews. One reference is here:Guidance for conducting systematic scoping reviews. Guidance for conducting systematic scoping reviews. International Journal of Evidence-Based Healthcare. 13(3):141–146, SEP 2015. Micah D.J. Peters; Christina M. Godfrey, DOI: 10.1097/XEB.0000000000000050

Response: Thank you for the reference. We have now modified the manuscript according to the guidelines provided.

Reviewer: 2

Reviewer Name: Sarah Prentice

Institution and Country: LSHTM UK

Please state any competing interests or state 'None declared': None Declared

Please leave your comments for the authors below

The authors have responded to my concerns and explained their methods more clearly in the revised manuscript. It is therefore a technically acceptable paper. However, I'm just not sure whether the results of this study would be of interest to many readers without an assessment of the quality of the studies they are reviewing (just listing possible available treatments and barriers is not that useful if some assessment of the extent to which they would be beneficial/a hinderance is not made at the sametime).

Response: The main aim of this review is to scope out the different guidelines and primary studies conducted for managing neonatal pneumonia in India, while also identifying the barriers to its case management. These findings will be integrated in a mixed-methods synthesis, with findings from a pan-India qualitative study, which is the ultimate aim of this funded project on neonatal pneumonia in India. We appreciate the importance and your recommendation of performing a quality assessment. We are performing a quality assessment of the guidelines as a separate activity, and will be using it while creating a policy brief, though we are unable to include it within the protocol of this review. We will also explore the scope for publishing the results of a quality assessment in the future.